# Improving Motor Skills in Early Childhood through Goal-Oriented Play Activity

**DOI:** 10.3390/children8110994

**Published:** 2021-11-02

**Authors:** Panggung Sutapa, Kukuh Wahyudin Pratama, Maziah Mat Rosly, Syed Kamaruzaman Syed Ali, Manil Karakauki

**Affiliations:** 1Department of Sport Science, Faculty of Sport Science, Yogyakarta State University, Yogyakarta 55281, Indonesia; 2Department of Sport Science Research and Development, Institute of Advanced Science, Engineering, and Education (IASEE), Shah Alam 40150, Malaysia; kukuh.pratama@iasee.org; 3Department of Physiology, Faculty of Medicine, University of Malaya, Kuala Lumpur 50603, Malaysia; maziahmr@um.edu.my; 4Department of Physical Education and Health, Faculty of Education, University of Malaya, Kuala Lumpur 50603, Malaysia; syed@um.edu.my (S.K.S.A.); manil@siswa.um.edu.my (M.K.)

**Keywords:** motor skills, childhood, goal-oriented play activity

## Abstract

Goal-oriented play activity encourages children to exercise and triggers the movement of body muscles. Active muscles are stronger, more agile, faster, and more powerful. Purpose: The purpose of this study is to determine that goal-oriented play activity in early childhood improves motor skills. Methods: Forty children aged 4.5–6 years old were recruited and took part in a set of training activities divided into Posts 1–5. To pass each post, a child had to run, walk on a balance beam, move sticks, throw and catch the ball, and arrange blocks. Training was given three times every week for 12 weeks. Data collected consisted of running 25 m, walking on the balance beam, throwing the ball as far as possible, locomotor movement, arranging cans, and bouncing the ball. Paired t and Wilcoxon signed-rank tests were used for analysis. Results: The results showed that there were significant differences in the motor skills evaluated pre- and post-training with *p* < 0.05. ***Conclusion:*** All six training activities conducted for 12 weeks provided significant improvements in the motor skills areas in children aged 4.5–6 years old.

## 1. Introduction

Play is a fun and exciting activity for children. Playing is a necessity, so it is suitable for people to say that playing is a child’s world. Playing provides an opportunity for children to adapt to others and the environment and impacts on their motor development, thinking ability, and the ability to solve problems [1]. There are still many parents who prioritize children’s academic skills in reading and mathematics, because they assume that being good at reading and math means having more abilities [2]. This kind of assumption is based on the presumption that play is a waste of time. In fact, many children can benefit from more targeted and planned play such as goal-oriented play activity.

Goal-oriented play is active play that requires working the body muscles, which stimulates muscle functions. Sufficient muscle activity makes the blood flow to the brain smoothly, increasing blood circulation. Motor movement is only possible when the muscular system attached to the bones and nerves innervates them. Enjoyable playing makes children happy and willing to repeat the same activity, so they do not feel that they are training the muscles to work. As stated by St. John [3], motor exercises and activities will benefit all students with delays, but the larger the delay, the more important the training and the bigger the impact on academics. Reduced physical activity in children can lead to reduced physical abilities, resulting in a decrease in motor quality [4]. Children who grow and develop with low motor abilities lose confidence. Adequate motor capabilities and nervous system development enable children to coordinate their limbs properly.

Early childhood is the golden age. It is the most important time to stimulate their motor abilities. The development of the nervous system begins at this time so the stimulation will greatly help accelerate the development of motor capabilities. Children who only play passively, such as playing with touch screen (on their smartphones), will experience a decrease in their motor skills [5]. The development of motor skills plays an important role in daily life so the development of motor skills should be optimal. Passivity in children has an impact on fat accumulation, contributing to obesity, while children who are actively moving have an increased chance of becoming more dynamic and healthier persons. From the explanation above, there are clearly two conclusions. The first is that playing helps children develop their motor skills. Second, spending more time at play will contribute to the child’s good physical health. Thus, this research will determine whether goal-oriented play activity can improve motor skills in early childhood.

Goal-oriented play is a form of play that is carried out through activities moving from one post to another continuously. It comprises different types of activities that must be accomplished sequentially. Mixed play makes it possible to develop motor elements such as strength, durability, agility, speed, balance, and coordination. Play is a necessity for some people because it can provide relief from the various problems they face. Bergen [6] stated that the key to play is fun; thus, the activities that do not provide pleasure cannot be called play. Furthermore, he explained that play ignites creativity and gives us pleasure. This is in line with what Mukherji and Dryden [7] stated, that play offers children the opportunity to develop key skills across domains. These skills associated with each developmental domain are social development, physical development, intellectual development, and communication/language development.

Play is an enjoyable activity that has an impact on pleasure and self-satisfaction. The activity is not based on age or gender, there is no coercion, and the default rules provide satisfaction for the player. It was supported by Sutapa et al. [8], who stated that play activity reflects the existence of the self, the part of the organism that exists both independently and interdependently and that can reflect upon itself and be aware of its own existence. Play for children who are still in the growing and developmental stages helps them to understand the outside world and to develop and stimulate their problem-solving abilities, which also increases their understanding of these life challenges. This is in accordance with Hong et al. [9], who argued that play offers trial and error activities that develop children’s intellectual intelligence. Play is performed by moving muscles repetitively to increase muscle strength, speed, and agility, resulting in the development of gross and fine motor skills.

The benefits of play for children include not only making them sweat but also facilitating growth and development, developing intra- and inter-personal intelligence, improving sense sharpness, and spurring creativity. Play can also be a therapeutic medium. Play supports children’s expression of fun and creative thinking, offering them a new experience in exploration. Play can also encourage children’s activity, as reflected in an active family [10]. Playing actively provides the opportunity to adapt to others and the environment and to strengthen the muscles. Active play has the paradoxical effect of increasing attention span and improving the efficiency of thinking and problem-solving. Two hours of active play per day may help reduce attention deficits and hyperactivity [11].

Playing is a process connected to sensory tools that aid development in children, both physically and psychologically. Active play is a means of training children to strengthen their muscles so that they can become stronger, faster, and more agile and have better motion control [12]. Muscles function as human motor mobility devices regulated by the nervous system, so it is often said that muscles and nerves are the functional units of motion systems in term of physiology.

Motor skill is the ability of the nervous system to control motion performance. Motor skills are divided into gross and fine motor skills. Bardid et al. [13] stated that the MOT 4–6 was designed to assess the gross and fine motor skills of preschool children (4–6 years old) and allows early identification of children with motor delay. Gross motor skills include locomotor, object control, and balance skills. Furthermore, Rudd et al. [14] stated that fundamental movement skills are often described more precisely as basic stability, object control, or locomotor movements involving fundamental motor skill (FMS) levels. In the end, the preschool period is an important indicator for the later participation of children in many sports activities. Kokštejn et al. [15] argued that achievement of sufficient gross motor skills involves the ability to perform single movements that use large muscles, while fine motor skills comprise ability in movements that require coordination between different organs, e.g., hands, eyes, arms, and limbs. Fine motor skills are related to coordination between eyes and hands, eyes and feet, or eyes, hands, and feet, as well as the ability to move fingers [16]. Madrona [17] stated that the aim of motor development is to achieve the control of one’s own body so that we can exploit all its possibilities of action. This development is shown through motor function, which consists of the rehearsal of movements aimed at the different relationships children establish with the world surrounding them. Payne and Isaacs [18] affirmed that fine movements are primarily regulated by the small muscles or muscle groups. Many movements performed by the hands are considered fine movements because the smaller muscles of the fingers, hand, and forearm are critical to finger and hand movement. Similarly, Johnston and Halocha [19] argued that fine motor skills are those manipulative skills that involve small movements and small muscles in acts such as picking up, feeding themselves, treading, drawing, cutting, and dressing. Fine motor skills develop slightly later than gross motor skills and require patience and practice to develop. Furthermore, Hill and Khanem [20] mentioned that motor development and its impact on other areas of physical and mental health as well as cognitive achievement are also central areas of focus for those working with children with neurodevelopmental disorders. In addition, Wang [21] stated that elements of motor capability include strength, durability, speed, agility, balance, and coordination. These elements of motion are basic elements in various sports that are indispensable to further development [22]. Physical activity cannot be separated from its three main elements: nerves, muscles, and bones. Nerves play the most important role in body activity supporting the muscles and bones, meaning that no movement occurs without the activity of the nerves. The two systems of muscles and bones support each other in the development of locomotor, non-locomotor, and manipulative motor skills. Locomotor skill is often interpreted as moving from one place to another, non-locomotor skill is undertaken without moving, and manipulative motor skill is movement when playing with a particular object.

Play is grouped into three categories, namely, sensorimotor, role-playing, and constructive. Sensorimotor play is the simplest form of play and is characterized by repeated muscle movements that increase strength, speed, agility, and balance. Role playing is spontaneous and constructive play where the main goal is to make the doer happy. Semoglou, Alevriadou, and Tsapakidou [23] stated that early childhood education emphasizes the significance of fine motor tasks and the use of learning aids, and Spanaki et al. [24] claimed that the fine motor training program had a positive effect upon the graph motor skills of kindergarten and early elementary school children. Chen et al. [25] explained that motor exercises and activities will benefit all students with delays, but the larger the delay, the more important the training and the bigger its impact on academics. Robinson et al. [26] explained that a mastery climate is an innovative and exceptional pedagogy for teaching children motor skills and encouraging physical activity. Koralek [27] found that manipulating and using tools with playdough and clay lets pre-schoolers discover the properties of this pliable material.

This research is needed to prove that goal-oriented play activity is very effective in improving motor skills in children. In addition to their enjoyment, children do not feel bored when playing. Each element in the different game posts strengthens the elements of their motor skills so that they complement each other. This research is focused on children aged 5–6 years old because this is a sensitive period/golden age; childhood is the period with the highest level of activity throughout life, so innovation is needed to optimize and facilitate early childhood physical activity. One alternative is through goal-oriented play activity for children aged 5–6 years. This is because the ages of 5–6 years are the best time to stimulate motor development. Our research uses this protocol to propose that goal-oriented play activity in early childhood can optimize children’s motor skills. The remaining sections of this paper are organized as follows: Section 2 reviews all materials and methods, Section 3 presents the results, Section 4 presents the discussions about this study and finally Section 5 concludes the findings with some direction for future work.

## 2. Materials and Methods

The research was approved by the ethical committee of the University of Yogyakarta (Ethics No: T/12/UN34.21/PT/2021). Forty participants were recruited from the children of Yayasan Ratih Kesuma Playgroup, consisting of 20 boys and 20 girls. The sample size was determined using an estimate of 30 individuals based on Cohen’s formulation [28] to achieve a power of 0.8, significance criterion α = 0.05, an effect size of 0.25 (large), and was within a large sample size population (population proportion = 0.5). Participants in this research were normal children aged 4.5–5 years old. The training study was needed because it helps to accelerate motor development. Motor skills play an important role in everyday life; therefore, the development of motor skills is one aspect of children’s development that must be optimized. The training study (goal-oriented play activity) data collection obtained permission from the school principal and all parents. Moreover, the activity was fun and safe. The game included six items. One game item lasted seven min, and each child had two chances. Thus, 7 min × 2 (two chances) × 6 game items = 84 min/day. This took place over 12 weeks, but on only 3 days per week, so the total was 36 days. The total duration was 84 min × 36 = 3024 min/50.4 h.

This training study program was carried out from 3 February to 3 April 2020 starting at 3:00 p.m. so it did not interfere with class hours. The training study was carried out in the community sports park, and the supervisor was the researcher himself assisted by the school principal. The researcher was assisted by two assistants when collecting the data. References indicating that the training study method of this program has a significant effect on improving motor skills are: (1) Wasenius et al. [29], who explained the effect of a physical activity training study on pre-schoolers’ fundamental motor skills; (2) Veldman, Jones, and Okely [30], who explained the efficacy of a gross motor skill training study in young children in an updated systematic review; and (3) Indah Lestari and Tri Ratnaningsih explained the effects of a modified game on the development of gross motor skills in pre-schoolers. Table 1 explains how each activity in the training study was performed [31].

To illustrate, Figure 1 below is an outline of the details of the pre- and post-data collection procedures and what units of measurement were used.

Below is the list of references for why such a measurement was used, what the significance of the measurement was in relation to fine motor skills improvement, and how it related.
Adpriyadi (2016). “Improving gross motor skills through the hopscotch traditional game for children of group B at TK Tunas Gading Yogyakarta in the 2015/2016 academic year” [32].Gümüşdağ, H. (2019). “Effect of pre-school play on motor development in children” [33].Moghaddaszadeh, A., and Belcastro, A. N. (2021). “Guided active play promotes physical activity and improves fundamental motor skills for school-aged children” [34].Khasanah and Sutapa (2018). “Development of teaching model through circuit game to increasing children’s gross motor ability” [35].

This measurement was carried out because it was in accordance with the elements that define motor skills, namely, the elements of speed, leg power, arm power, and balance of coordination between eyes and hands. This measurement was very appropriate for the rating of motor skills because the elements of motor skills are speed, power, balance and coordination, and the skills increase after training. Because there is an increase in the elements or parts, there will be an overall increase. Below are the instruments/equipment used when data were collected:To run, the necessary tools were a running track and a stopwatch.The long jump took place without prior running (prefix), and the tool required was a yardstick.The ball was thrown as far as possible, and the tools required were a tennis ball and yardstick.The coordination test required arranging the cans into a triangle shape from highest to lowest, 5 then 4, then 3, 2, 1; the tools needed were used cans and a stopwatch.The tools needed for balance were a walkway beam and a stopwatch.The coordination exercise required throwing and catching the ball, and the tools needed were a tennis ball and a stopwatch.

Before the training program took place, the children were instructed about the activities to be carried out, namely, playing to improve their motor skills, but because of their nature the children often asked questions. The children were informed about the results of this data collection, and they were also told that their motor skills had improved, which made them happy.

The research involved 40 participants consisting of 20 boys and 20 girls aged 4.5 to 5 years. Participants were presented with goal-oriented play tasks moving from post 1 to 5 and comprised of walking on a balance beam, moving sticks, jumping goalposts, throwing balls, and arranging blocks. Movement from post 1 to 5 was carried out by running. This training was performed 3 times a week for 12 weeks. The data were collected using the 25 m run for the speed test, walking on a balance beam for the balance test, throwing the ball as far as possible to measure explosive arm force, the locomotor jump test to measure explosive limb power, arranging cans for eye–hand coordination, and bouncing the ball to measure eye–arm coordination.

The data were assessed for normality using Kolmogorov–Smirnov Z and Shapiro–Wilk tests. The results indicated that data for throwing the ball and bouncing the ball were consistently significant at *p* < 0.05 for both the pre- and post-training study measures. Items for the 25 m run, walking on a balance beam, locomotor jump test, and can arranging were normally distributed with *p* > 0.05 for the pre-test but not normally distributed in post-test for the 25 m run and balance beam. Paired t tests were used for normally distributed data and Wilcoxon signed-rank tests were used for data that were not normally distributed. Pre- and post-data were compared for improvements following the 12-week training study program, and the significance level was set at *p* < 0.05 at a 95% confidence interval.

## 3. Results

The results showed that there were differences between pre- and post-training in the elements of motor ability in early childhood with a significance level of *p* < 0.05. For each component of motoric ability, there were differences in the 25 m run, locomotor jump, ball throws, can arrangement, and ball bounce. The 25 m run measured speed, the locomotor jump measured explosive power, ball throw as far as possible measured the arm’s explosive force, can arrangement measured eye and hand coordination, walking on a balance beam measured balance, and ball bounce measured the eye–hand coordination.

From Table 2 below, we see that all participants increased their ability between pre-training and post-training. Table 3, Table 4 and Table 5 summarizes the results of the pre and post-training improvements.

## 4. Discussion

During childhood and at pre-school age, movement is an integral part of children’s lives. In the first six years of life, children discover themselves and the world through movement and capture their surroundings through their body and their sensations [36]. Thus, especially in that period of human life, the study of a child’s motor performance can significantly contribute to the full understanding of his/her entire personality [37]. Such a study is considered essential to the better preparation of children in the field of learning and for preventing the development of motor disorders associated with a sedentary life and lack of motor skill training during the period of growth and development [38]. Additionally, the sound assessment of a child’s motor development level is directly associated with the planning of developmentally adequate movement programs.

One of the most significant early childhood movement training programs involves goal-oriented play activities. This form of play involves repeating activities related to a specific skill set over a series of training durations. Playing usually involves physical activities that are considered fun, in accordance with the nature of children to move in various ways during play. Play that is perceived to be “fun” tends to engage and motivate children to perform repetitive motions or activities while adhering to the routine over a much more prolonged period of duration. Children do not feel that doing the same activity is training their muscles to improve their motor skills. Conversely, the lack of physical activity can lead to the decrease in their motor skills [39].

In this study, goal-oriented play consisted of a series of activities comprised of walking on a balance beam, moving sticks, jumping goalposts, throwing balls, and arranging blocks. The results of this research showed that there were significant differences (*p* < 0.05) before and after the training was given. This showed that goal-oriented play can improve motor skills in early childhood for children aged 4–6 years old. This was in accordance with Altinkök’s [40] research that showed that goal-oriented play can develop motor elements such as strength, durability, agility, speed, balance, and coordination. Goal-oriented play can develop multilateral skills, helping to build basic abilities. Basic movement, such as walking on the balance beam, moving sticks, jumping goalposts, throwing balls, and arranging blocks, is a part of children’s educational and learning experiences that can be implemented easily in day care or preschool. The skills learned in this period will be permanent and will provide the basis for new skills [41]. Early childhood motor skill training can prove beneficial for physical literacy into adulthood, as these aspects contribute to forming independent adulthood [42].

This age bracket (4–6 years old) is considered the golden age to improve children’s gross and fine motor skills, so this is the right time to maximize their motor development [43]. Active play is characterized by repetitive muscle movement, which increases strength, speed, agility, and muscle flexibility. Its effect is the improvement in children’s motor abilities. The six elements are the 25 m run, walking on a balance beam, throwing the ball, the locomotor jump test, arranging cans, and bouncing the ball, which are aspects that, when included in continuous training, improve the child’s motor skills. With these six elements, if given in training or carried out continuously, muscle adaptation occurs, increasing muscle ability in children.

The muscles that are given training will have thickened myelin, so there will be an increase in muscle endurance. Muscles will experience increased ability and myelin thickening, if given training for at least 12 weeks with a training frequency of at least three times a week. This program is the shortest compared to other programs but has a significant effect. Implementing a similar training method, based on larger scales (i.e., a larger number of children and with multi-site programs) has the potential to maximize their motor ability. This avenue will lead to direct and indirect improvement in children’s motor skills, which can eventually translate into talent creation. Talent identification is crucial during early childhood to maximize duration availability for honing fine and gross motor skills as early as possible. After training in these six elements, children’s motor abilities will increase; of course, with the increase in motor abilities, the children will become more agile, stronger, and quicker to act and have better coordination. Play with a sensorimotor approach requires children to learn to use tools that are adapted to the surrounding conditions, so that active muscles carrying out motion are stimulated. The development of motor skills for children is very important because motor ability is the foundation of daily life. Motor ability involves the ability of the body muscles to perform activities such as walking, running, and jumping [44]. Lopes et al. [45] argued that children’s motor skill development incorporates many body systems, including sensory, musculoskeletal, cardiorespiratory, and neurological systems.

Motor ability is necessary for movement. For a person to move effectively and efficiently, he or she must possess physical abilities such as strength, agility, speed, flexibility, coordination, and balance. Cohen et al. [46] stated that participation in physical activity is vital for enhancing children’s physical, social, cognitive, and psychological development. Higher levels of physical activity in children are associated with improved fitness (both cardiorespiratory fitness and muscular strength), enhanced bone health, and reduced body fat. Similarly, Pahlevanian and Ahmadizadeh [47] affirmed that motor skills play an important role in children’s learning and improve the growth of other important learning skills such as educational and social cognition. Meanwhile, Fallah, Nourbakhsh, and Bagherty [48] reported that physical movement is one of the most important aspects of human life, and motor skills allow children to gain greater control over their living environment. Systematic physical activity training can improve motor skills, resulting in changes in a person’s physical performance and endurance. Regular physical activity can also help improve quality of life by reducing risks from the onset of secondary diseases (i.e., diabetes mellitus, hyperlipidemia, heart diseases) that are associated with a sedentary lifestyle [49]. Additionally, motor skill competence was strongly associated with physical activity levels in children within this age group but may differ between genders according to the intensity of physical activities performed [50]. Regular muscle activity improves blood circulation so that the substances needed by the nervous system and muscles will be delivered. The impact of fulfilled nutrients on the nervous system and muscles makes a person fitter, by increasing speed, endurance, balance, and coordination, which contribute to motor ability.

For future research, it will be necessary to develop research into increased multiple intelligence consisting of mathematical logic, language, intrapersonal, interpersonal, art/music, spatial, kinaesthetic, and spiritual abilities through goal-oriented play activities in early childhood. By developing and implementing this research, we hope to optimally improve children’s motor development.

However, there are some limitations in this research. First, this method is only applicable to normal children in the day-care environment with routine monitoring after the school schedule; this method has not been tried with the older children who already attend school. Second, this method had an impact because it involved mentors. This method has not been tried with self-regulated training, such as home-based training, to see if it will have the same effective impact. Lastly, this method focused only on psychomotor behaviour or neuromuscular behaviour, and it cannot measure the impact on cardiovascular, strength duration, and endurance abilities.

## 5. Conclusions

Goal-oriented play activity is fun for children because it makes them eager to repeat the same activity, and many benefits are gained through active play. The muscles’ ability to move will increase, and since they are stimulated, the child’s motor ability will also increase. Goal-oriented play is divided into five posts, namely, walking on the balance beam, moving sticks, locomotor jumping, throwing balls, and arranging blocks and can improve the motor skills in early childhood.

All six training exercises in the 12-week training program provided significant improvements in the motor skills areas of children aged 4.5–6 years. The level of improvement in motor skills during early childhood may benefit from goal-oriented active play and increased motor ability.

## Figures and Tables

**Figure 1 children-08-00994-f001:**
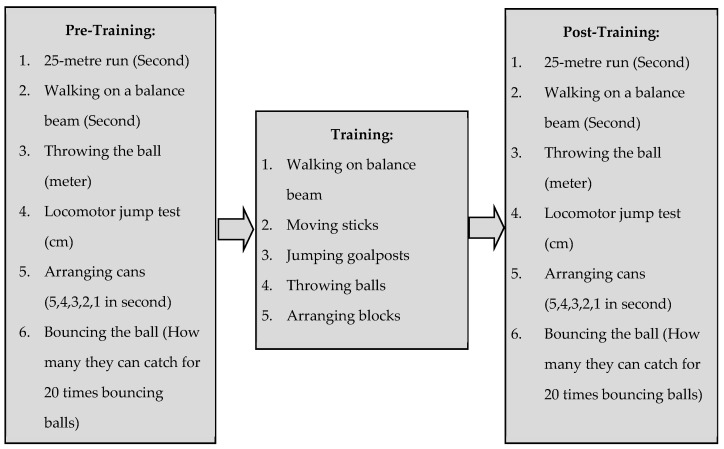
Pre- and post-data collection procedures.

**Table 1 children-08-00994-t001:** The training study.

No	Activity/Training	Description
1	Walking on balance beam	Starting point as POS 1, after finish run to POS 2
2	Moving sticks	This is POS 2, after finish run to POS 3
3	Jumping goalposts	This is POS 3, after finish run to POS 4
4	Throwing balls	This is POS 4, after finish run to POS 5
5	Arranging blocks	This is POS 5, the finishing point

**Table 2 children-08-00994-t002:** Demographic profiles of participants.

Demography	Boy	Girl	Combined
Sample size	*N* = 20	*N* = 20	*N* = 40
Age (years)	5.96(0.39)	5.83(0.38)	5.9(0.39)
Height (cm)	113.75(1.40)	112.33(2.04)	113(1.87)
Weight (kg)	20.24(0.82)	19.23(1.39)	19.73(1.24)
BMI	15.64(0.67)	15.24(1.15)	15.44(0.95)

**Table 3 children-08-00994-t003:** Descriptive data per training.

Training Type	Pre-Training Values	Post-Training Values
25 m run (s)	9.71(1.76)	4.98(1.03)
Throw the ball (m)	* 2.76(0.98)	* 5.90(1.72)
Locomotor jump (cm)	46.80(7.60)	78.28(13.10)
Can arrangement (s)	11.14 (1.62)	7.18(1.01)
Balance beam walk (s)	9.83(1.49)	6.89(0.85)
Throwing and catching the ball (count/20 times)	* 11.0(4.0)	* 16.5(2.75)

S: second; m: meter. All values are in mean (standard deviation) unless otherwise stated. * Indicates the values are given in median (interquartile range).

**Table 4 children-08-00994-t004:** Paired *t* test between pre- and post-training study of 25 m run, locomotor jump, can arrangement, and balance beam walk.

Training Study	Mean (Standard Deviation)	95% Confidence Interval		Significance Level (*p* Value)
		Lower Limit	Upper Limit	
25 m run (second)	4.74(1.68)	4.20	5.27	0.000
Locomotor jump (cm)	−31.47(12.61)	−35.61	−27.44	0.000
Can arrangement (second)	3.96(1.39)	3.52	4.41	0.000
Balance beam walk (second)	2.94(1.45)	2.47	3.40	0.000

**Table 5 children-08-00994-t005:** Wilcoxon signed-ranks test for pre- and post-training study of throwing the ball and of throwing and catching the ball.

Training Study	Mean Rank	95% Confidence Interval		Significance Level (*p* Value)
		Lower Limit	Upper Limit	
Throw the ball (meter)	39	0.000	0.000	0.000
Throwing and catching the ball (count/20 times)	20.50	0.000	0.000	0.000

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
