# Peer review of "Improving Motor Skills in Early Childhood through Goal-Oriented Play Activity"

_children, 2021, doi:10.3390/children8110994_

Round 1

Reviewer 1 Report

Dear authors,

you elaborate on a really interesting topic, especially in the covid19 era, where children were sitted all day in front of a screen. 

Play helps children to grow in every aspect, so it is a necessity to develop all necessary motor skills to facilitate children to play.

I have to admit that I struggled to understand your paper. You need to rephrase almost everything in your paper in proper English. 

You must use more citations. For example, in the introduction (lines 31-35) you write about "common" parents and their beliefs about play.  First of all, what do "common" parents stand for? Secondly, where is the citation? Is there research that proves this belief? 

Why continuous play is active play? Children can play, without being active.

And many, many things. They are all over your paper.

You repeat your sample twice. At first, they are boys and girls and then males and females. 

You have incorporated in your intervention, activities that they are also items of your test. This is not intervention. This is training. In my opinion this a huge methodological issue. If your purpose was training, then you are safe. What is the case? Training, treatment or intervention? You know.

Please, rewrite your paper. Read again your references, search for more, citate your paragraphs, rewrite your methods, link better your results with literature in your discussion.

Thank you for giving me the opportunity to review your paper. Good luck.

Author Response

Dear Reviewer 1,

Thank you very much for your time to review our article.
We have done with the revision according to your comment.

Comments

Answer and Action

You elaborate on a really interesting topic, especially in the covid19 era, where children were sitted all day in front of a screen.

Thank you very much!

Play helps children to grow in every aspect, so it is a necessity to develop all necessary motor skills to facilitate children to play.

Thank you very much!

I have to admit that I struggled to understand your paper. You need to rephrase almost everything in your paper in proper English.

Edited and proofread by MDPI proofread service

You must use more citations. For example, in the introduction (lines 31-35) you write about "common" parents and their beliefs about play.  First of all, what do "common" parents stand for? Secondly, where is the citation? Is there research that proves this belief?

Done added

Why continuous play is active play? Children can play, without being active.

Adjusted to goal oriented play

And many, many things. They are all over your paper.

Noted and adjusted

You repeat your sample twice. At first, they are boys and girls and then males and females.

Done adjusted

You have incorporated in your intervention, activities that they are also items of your test. This is not intervention. This is training. In my opinion this a huge methodological issue. If your purpose was training, then you are safe. What is the case? Training, treatment or intervention? You know.

Done adjusted to “training”

Please, rewrite your paper. Read again your references, search for more, citate your paragraphs, rewrite your methods, link better your results with literature in your discussion.

Done added

Thank you for giving me the opportunity to review your paper. Good luck.

Thank you very much!

Please see our revised article in the attachment.
This article has been proofread by MDPI team as well.
Thank you!

Regards,
Sutapa

Author Response

Dear Reviewer 2,

Thank you very much for your time to review our article.
We have done with the revision according to your comments.

Comments

Answer

Line 56: while those who “are”

Done adjusted

Lines 61-62: Reword this sentence as it is unclear what is being said.  

Done adjusted

Line 69: Further, he “explains”

Done adjusted

Line 70: Play should not be capitalized.

Done adjusted

Line 75: Back part of this sentence needs to be reworded, used “and” twice without clear flow.

Done adjusted

Line 94: Two “hours” of active play?

Done adjusted

Lines 138-139: Manipulative skills definition needs to be more clearly worded.

Done adjusted

Lines 147-148: “John (34) motor”, this does not make sense, reword.

Done adjusted

Line 149: When citing research, the citation should come at the end rather than the beginning unless worded correctly.

Done adjusted

Line 154: Does not make sense starting at “Besides”?

Done adjusted

Line 269: continuous “play”

Done adjusted

Line 272: “Play” that “is” perceived

Done adjusted

Many of the wording in the discussion is almost complete repetition of the introduction. Writing needs to be more diverse and leading to a purpose instead of just restating information.

Done edited the discussion

Lines 277-278: The practice skills done can not be listed as games, as the intent of the tasks never had a game focus. Also, I question the definition of play and the structure of the intervention/program.

Done adjusted

Line 287-288: To state these skills learned will be lifelong permanent, research must be cited here. The data you provided does not back this statement.

Done added

Line 344: Wasn’t this study more on psychomotor behavior or neuromuscular behavior rather than cognitive behavior?

Done adjusted

Lines 349-352: Continuous play as defined by this study?

Done adjusted

Provide more descriptive data in the results section (e.g., average scores for each category).

Done added

The title states continuous play but the prescribed program implemented does not sufficiently fit the definition of continuous play. I would consider renaming the article.

Done adjusted

The literature review highlights the importance of play to decision making and problem solving, which is true. But the type of “play” that is done in this study would not drive those findings. This is much more of a controlled regimen than actual play. I am struggling with the definitions and the actual implementation. They do not seem to line up. It seems as though this is more of continuous motor skill tasks that then improve motor skills, which is important.

Done adjusted with goal oriented play term

The paper needs to be proofread for English grammar rules and flow. Many areas of the writing are choppy and there are a multitude of grammatical errors.

Edited and proofread by MDPI proofread service

The research does have value in providing a narrative that kids need to develop motor skills through movement, and they definitely can. But the wording and approach need to be adjusted to develop a more accurate and compelling argument.

Thank you, this article will be proof read by MDPI service

Please see our revised article in the attachment.
This article has been proofread and edited by MDPI team as well.
Thank you!

Regards,
Sutapa

Round 2

Reviewer 1 Report

You have made many changes, so now your article looks so much better. Keep in mind both reviewers  comments' for future designs.

Author Response

Dear Prof Reviewer 1,

Thank you very much for your great advise.
Here I send the updated article after minor revision.
Please check it.
Thank you very much!

Regards,
Panggung

Reviewer 2 Report

Line 30: Does research say most parents assume that children who like playing are weak and stupid? If not, delete this sentence.

Well done on revisions, solid paper that highlights the need for motor skill development. 

Author Response

Dear Prof Reviewer 2,

Thank you very much for your great advise on our article.
Here I send the latest version of our article after minor revision.
We deleted the sentence "children who like playing are weak and stupid".
Please check it.
Thank you very much!

Regards,
Panggung
